# Targeting Inflammation and Oxidative Stress to Improve Outcomes in a TNBS Murine Crohn’s Colitis Model

**DOI:** 10.3390/nano14100894

**Published:** 2024-05-20

**Authors:** Anisha Apte, James R. Bardill, Jimena Canchis, Stacy M. Skopp, Tobias Fauser, Bailey Lyttle, Alyssa E. Vaughn, Sudipta Seal, David M. Jackson, Kenneth W. Liechty, Carlos Zgheib

**Affiliations:** 1Laboratory for Fetal and Regenerative Biology, Department of Surgery, University of Arizona Tucson College of Medicine, Banner Children’s at Diamond Children’s Medical Center, Tucson, AZ 85721, USAkliechty@arizona.edu (K.W.L.); 2Department of Surgery, School of Medicine, University of Colorado Denver, Anschutz Medical Campus, Aurora, CO 80045, USA; james.bardill@cuanschutz.edu (J.R.B.);; 3Advanced Materials Processing and Analysis Centre, Nanoscience Technology Center, University of Central Florida, Orlando, FL 32827, USA; 4Ceria Therapeutics, Inc., Tucson, AZ 85721, USA

**Keywords:** inflammatory bowel disease, Crohn’s disease, TNBS, colitis, nanomedicine

## Abstract

Inflammation and oxidative stress are implicated in the pathogenesis of Crohn’s disease. Cerium oxide nanoparticle (CNP) conjugated to microRNA 146a (miR146a) (CNP-miR146a) is a novel compound with anti-inflammatory and antioxidative properties. We hypothesized that local administration of CNP-miR146a would improve colitis in a 2,4,6-Trinitrobenzenesulfonic acid (TNBS) mouse model for Crohn’s disease by decreasing colonic inflammation. Balb/c mice were instilled with TNBS enemas to induce colitis. Two days later, the mice received cellulose gel enema, cellulose gel with CNP-miR146a enema, or no treatment. Control mice received initial enemas of 50% ethanol and PBS enemas on day two. The mice were monitored daily for weight loss and clinical disease activity. The mice were euthanized on days two or five to evaluate their miR146a expression, inflammation on histology, and colonic IL-6 and TNF gene expressions and protein concentrations. CNP-miR146a enema successfully increased colonic miR146a expression at 12 h following delivery. At the end of five days from TNBS instillation, the mice treated with CNP-miR146a demonstrated reduced weight loss, improved inflammation scores on histology, and reduced gene expressions and protein concentrations of IL-6 and TNF. The local delivery of CNP-miR146a in a TNBS mouse model of acute Crohn’s colitis dramatically decreased inflammatory signaling, resulting in improved clinical disease.

## 1. Introduction

Inflammatory bowel disease (IBD) describes a spectrum of debilitating chronic inflammatory disorders of the gastrointestinal tract, including Crohn’s disease and ulcerative colitis, and affects approximately five million individuals worldwide [1]. Despite advances in medical and surgical therapies, the prevalence of IBD has been on the rise, and its associated morbidity and mortality remains high. From 1990 to 2019, the number of IBD-attributed deaths rose by 68.75% [1]. Among IBD, Crohn’s disease has no cure, and many of the therapies available are limited by their cost, availability, and efficacy. For example, TNF blockers are newer systemic treatments for IBD, but approximately 30% of patients with Crohn’s disease do not respond to these medications and among those who do initially respond, half eventually develop tolerance [2,3]. Additionally, like many other available IBD therapies, TNF blockers are complicated by poorly tolerated side effects associated with their systemic delivery [4]. The highly complex and multifactorial pathophysiology of IBD has made the targeted treatment of Crohn’s disease challenging, however, recent advances in our understanding of associated molecular pathways may guide the development of novel therapeutics.

There are multiple pro-inflammatory signaling pathways implicated in the pathogenesis of Crohn’s disease. One of these is that of the nuclear factor kappa-light-chain enhancer of activated B cells (NF-κB) [5]. When activated, NF-κB translocates into the cell nucleus and transcribes several proinflammatory genes, including cytokine interleukin-6 (IL-6) and tumor necrosis factor (TNF). Activation of the NF-κB pathway can also induce the production of reactive oxidative species (ROS) and cell apoptosis. Physiologically, NF-κB activation occurs in response to pathogen-associated molecular patterns (PAMPs) or damage-associated molecular patterns (DAMPs), which indicate the presence of microorganisms or cell damage. The release of pro-inflammatory mediators in these scenarios leads to immune cell chemotaxis to destroy or defend against invading organisms and to repair cell damage [6]. In normal physiology, this reaction subsides upon resolution of infection or cell damage. In Crohn’s disease, a dysregulated adaptive immune response leads to inappropriate T-cell infiltration of colonic tissue in the absence of PAMPs and DAMPs, and the unregulated activation of NF-κB, resulting in epithelial inflammation and destruction [7]. Even with resolution of the inciting factor, a lack of regulatory feedback leads to a persistent state of inflammation [8].

ROS have been demonstrated to be as much of a pathogenic factor in Crohn’s disease as they are a byproduct of inflammation [9]. In Crohn’s disease, persistent inflammation leads to excess ROS production that overwhelms physiologic antioxidant defense mechanisms. The imbalance of production and elimination of ROS creates oxidative stress and can result in additional cellular and molecular damage [10]. The upregulation of ROS in peripheral mononuclear cells and colonic mucosa in patients with active Crohn’s disease has been demonstrated [11], as have reduced levels of serum-free thiols, which are a major component of the antioxidant machinery [12]. Certain enzymes responsible for the synthesis of ROS, such as inducible nitrous oxide synthase (iNOS), have also been shown to be upregulated in the colonic mucosa of IBD patients [13]. iNOS is responsible for the generation of nitric oxide (NO), a free radical implicated in epithelial injury, apoptosis, and the perpetuation of inflammatory responses [14]. As implied by its name, iNOS production can be induced by NF-κB activation [15], and its upregulation likely contributes to the imbalance of ROS and antioxidants. Multiple trials investigating the dietary effects of antioxidants in IBD patients have shown that a reduction in ROS correlates to reductions in inflammatory cytokines, but none have shown a reduction in clinical disease [10].

The only therapeutic currently used in the management of IBD that targets ROS is 5-aminosalicylic acid (5-ASA), an ROS scavenger delivered per rectum. 5-ASA has been shown to reduce disease activity in patients with ulcerative colitis but not Crohn’s disease [16]. Most current medical therapies for Crohn’s disease are designed to reduce disease activity through immune suppression. For example, steroids are a first-line treatment for IBD patients presenting with acute colitis and work by interfering with NF-κB activation in the nucleus of immune cells. Methotrexate, an immunomodulator that works by preventing T-cell activation and proliferation, and infliximab, a biologic medication that targets and inhibits TNF, are used to maintain remission in chronic disease [17]. Given the synergistic relationship between inflammatory cytokines and ROS, novel therapeutics would likely benefit from harnessing both anti-inflammatory and antioxidative mechanisms.

We previously developed a novel compound comprising a cerium oxide nanoparticle (CNP) conjugated to microRNA 146 (miR-146a) that we refer to as CNP-miR146a. CNP-miR146a contains multiple strands of microRNA chemically conjugated to a single CNP at the 3′ end via an amide linker through the use of 1,1′-carbonyldiimidazole chemistry. Cerium oxide is a metal oxide that exists in two oxidative states and can regenerate without energy on a nanoscale, allowing it to mimic the ROS-scavenging capabilities of the enzymes superoxide dismutase (SOD), catalase, glutathione peroxidase, photolyase, and phosphatase [18,19,20,21]. MicroRNA are small non-coding RNAs that have been found to play regulatory roles in cell signaling and have recently gained attention for their role in the pathophysiology of IBD [22,23,24]. MiR146a has been shown to directly downregulate NF-κB translocation into the cell nucleus by preventing the transcription of upstream activator proteins IRAK1 and TRAF6 [25]. MiR-146a works as a molecular break in inflammation produced by the NF-κB pathway and has been demonstrated to have a potential role in the pathophysiology of IBD [24]. CNP-miR146a contains both anti-inflammatory and antioxidant properties, which we have previously demonstrated in murine models for diabetic wounds and acute lung injury [26,27,28]. We, therefore, hypothesized that the administration of CNP-miR146a could similarly target inflammation and oxidative stress and improve clinical disease in a murine model for Crohn’s disease.

We tested our hypothesis in a well-established animal model for acute colitis that utilizes 2,4,6-Trinitrobenzenesulfonic acid (TNBS). When administered as an enema, TNBS haptenizes colonic proteins and triggers a predominant Th1 response of the mucosal immune system, with subsequent NF-κB activation and proinflammatory cytokine and ROS production [29,30,31]. The Th1 response mimics that of Crohn’s disease, making this model particular useful for studying acute flares of Crohn’s colitis [32]. We first investigated the relationship of miR146a expression to that of pro-inflammatory cytokines IL-6 and TNF in the colons of mice induced with TNBS colitis. We then demonstrated the successful use of a hydroxyethyl cellulose gel with CNP-miR146a to deliver miR146a to the colon. Subsequently, we examined the effects of cellulose gel with CNP-miR146a enema on clinical disease activity, colon tissue damage, and inflammatory cytokine and ROS production in the colons of mice induced with TNBS colitis.

## 2. Materials and Methods

### 2.1. Development of CNP-miR146a

CNP-miR146a was synthesized as previously described [33]. In brief, CNPs were synthesized via chemical hydrolysis, and then conjugated to miR146a using 1′-1′ carbonyldiimidazole (CDI) chemistry to covalently bind the miR146a amino group to the CNP hydroxyl group. The CNP-miR146a concentration was assessed by Quant-iT™ (ThermoFisher Quant-iT™ microRNA Assay Kit, Waltham, MA, USA) to determine the miR146a concentration, and inductively coupled plasma mass spectrometry (ICP-MS) to determine the cerium concentration. The conjugated CNP-miR146a was diluted in sterile phosphate buffer saline (PBS) to a concentration of 11.4 ng/uL and stored at −20 °C.

### 2.2. Development of CNP-miR146a Hydroxyethyl Cellulose Gel

A 2% hydroxyethyl cellulose gel was synthesized from a solution containing 0.17% methyl paraben, 0.02% propyl paraben, 5% glycerin, 0.263% sodium phosphate monobasic, 0.044% sodium phosphate dibasic, 3% propylene glycol, 0.05% EDTA, 0.05% d-mannitol, and 2% hydroxyethyl cellulose (MW 1,300,000) dissolved in sterile phosphate buffer (PBS) and combined with CNP-miR146a for a final therapeutic concentration of 0.125 ng/uL. This solution was stirred with a magnetic stir bar for approximately 60 min at room temperature until reaching a gel-like consistency.

### 2.3. TNBS Acute Colitis Animal Model

All animal studies were approved by the Institutional Animal Care and Use Committee (IACUC, protocol 2022-0923) at the University of Arizona College of Medicine. Animal care was performed by trained veterinarians and technologists according to the NID Guide for the Care and Use of Laboratory animals. All mice were maintained in standard housing for at least 48 h prior to experimentation to allow for acclimation.

Six- to eight-week-old Balb/c male mice (Jackson Laboratory, Bar Harbor, ME, USA) weighing at least 19 g were randomly assigned to receive an enema of 2.5% TNBS or 50% ethanol. A 2.5% solution of TNBS was made from the original 5% TNBS (Sigma-Aldrich SKU P2297, St. Louis, MO, USA) stock by diluting it to 50% with 200 proof ethanol. The 2.5% TNBS was administered at a dose of 75 mg/kg. The control mice received an equivalent volume of 50% 200 proof ethanol diluted with RNase free water. Prior to enema administration, all mice were weighed and placed in a ventilated prestaging box to encourage spontaneous evacuation of their bowels. Enemas were administered under isoflurane anesthesia using a lubricated plastic 22-gauge × 38 mm feeding tube (Instech Lab, Plymouth Meeting, PA, USA), which was advanced its entire length past the anus. Following instillation of the enema, mice were maintained in a head-down position for at least 30 s to prevent leakage.

Following their recovery from anesthesia, the mice were placed in single-cohort housing with unrestricted access to food and water, including moist chow. The mice were weighed and assessed for symptoms of disease, including bloody stools, diarrhea, and fatigue, as well as for symptoms of dehydration by evaluating skin turgor, mucous membranes, and body temperature. The mice were rehydrated as necessary by replacing 20% of the total blood volume of the animal, estimated as 6% of their total body weight, with normal saline administered intraperitoneally, as previously described [34].

Two days after the initial enema, the mice that demonstrated at least 10% weight loss in combination with the presence of bloody stools or diarrhea were determined to have developed colitis and were randomized to either continue their course untreated, receive a cellulose gel enema, or receive a cellulose gel with CNP-miR146a enema under anesthesia. All control mice received a PBS enema under anesthesia. The same technique was used for the administration of these enemas as previously described. All mice were returned to their housing following the procedure and continued to receive the same post-procedural care and monitoring. At either two or five days following the initial enema administration, the mice were euthanized using carbon dioxide, and colon tissue was collected for analysis. For the two-day study, the untreated TNBS mice were euthanized at 48 h following delivery (*n* = 6), and the TNBS mice that received cellulose gel with CNP-miR146a enemas were euthanized at either six (*n* = 6) or twelve (*n* = 5) hours after the therapeutic enema. Four cohorts were used in the five-day study: controls (ethanol + PBS enema, *n* = 8), untreated (TNBS + no enema, *n* = 10), cellulose gel (TNBS + cellulose gel enema, *n* = 7), and cellulose gel with CNP-miR146a (TNBS enema + cellulose gel with CNP-miR146a enema, *n* = 10).

### 2.4. Clinical Assessment of Acute Colitis

The mice weights were taken daily for the calculation of the percentage of weight loss. The weights were taken prior to enema administration or rehydration to minimize confounding.

### 2.5. Histopathological Studies

A lengthwise section of the colon was made into a Swiss roll and stored in a histology cassette in 10% formaldehyde. This was kept at room temperature for 24–48 h before dehydrating in a 70% ethyl alcohol and stored at 4 °C. The dehydrated tissue was embedded in paraffin blocks and sectioned at 4 μm. The mounted sections were deparaffinized and stained with hematoxylin and eosin. The tissue sections were viewed and photographed at 20× and 100× using brightfield microscopy with a digital Keyence BZ-X microscope (Itaska, IL, USA) and scored with a previously designed scoring system for chemical-induced colitis [35]. Briefly, a score of 1–3 is assigned based on the severity and extent of inflammatory cells within each tissue layer. A value of 1 is assigned to specimens showing mild inflammatory infiltration of the mucosa. A value of 2 is assigned to specimens showing moderate inflammatory infiltration of the mucosa or inflammatory cell infiltration of the submucosa. A value of 3 is assigned to specimens that show inflammatory infiltration penetrating the submucosa, indicating transmural inflammation. A second score of 1–3 is assigned based on changes to the mucosal architecture, with a score of 1 assigned for the presence of focal erosions only, a score of 2 assigned for moderate ulcerations, and a score of 3 assigned for extended ulcerations with or without granulation tissue or pseudopolyps. The sum of these two sets of scores is used to calculate a histology injury score (Table 1).

### 2.6. RNA Isolation and Quantitative Real-Time PCR (qRT-PCR)

Colon tissue was collected from animals euthanized either two or five days after the initial enema instillation. The colon was instilled with PBS with the expression and removal of fecal contents. The colon was split into three lengthwise sections, with two sections snap-frozen in liquid nitrogen and stored at −80 °C. The total RNA was extracted from frozen colon tissue using qiazol (Qiagen, Hilden, Germany) and mechanical homogenization according to the manufacturer’s instructions. The total RNA was quantified using a Synergy LX multi-model plate reader (BioTek, Winooski, VT, USA) and treated with a DNAse kit (Invitrogen, Thermo Fisher, Waltham, MA, USA) prior to undergoing a reverse transcriptase reaction and conversion to cDNA with the High-Capacity cDNA Reverse Transcription Kit (Applied Biosystems, Thermo Fisher, Waltham, MA, USA). QRT-PCR was performed with the CFX96 Touch Real-Time PCR Detection System (Bio-Rad, Hercules, CA, USA) using PrimeTime Gene Expression Master Mix (Integrated DNA technologies, Coralville, IA, USA) to evaluate the relative gene expressions of proinflammatory genes IL-6 (ThermoFisher Mm00446190_m1 IL6), TNF (ThermoFisher Mm00443258_m1 Tnf), and inducible nitrous oxidase 2 (iNOS2) (ThermoFisher Mm00440502_m1 Nos2) using Glyceraldehyde 3-phosphate dehydrogenase (GAPDH) (ThermoFisher Mm99999915_g1 gapdh) as an endogenous control. For the evaluation of miRNA, isolated RNA was diluted to 5 ng/uL and converted into miR146a cDNA (ThermoFisherRT&TM snRNA 000468) and endogenous control U6 cDNA (ThermoFisher RT&TM snRNA 001973) using TaqMan microRNA reverse transcriptase kit (Applied Biosystems, ThermoFisher) and amplified using qRT-PCR as described. All qRT-PCR data were normalized to the five-day control mice.

### 2.7. Protein Assessment Using ELISA

The second set of snap-frozen colon tissue was used for protein analysis. The samples were homogenized in a lysis buffer made from RIPA lysis and extraction buffer (ThermoFisher Cat# 89900) and protease inhibitor with mechanical homogenization and sonication. The samples were centrifuged, and the supernatant was collected and quantified for the total protein concentration using a Pierce BCA assay (ThermoFisher Cat# A53225). The IL-6 and TNF protein concentrations were detected by ELISA (R&D Systems™ Mouse IL-6 DuoSet ELISA kit and Mouse TNF-a DuoSet ELISA kit, Minneapolis, MN, USA) with a standardized loading of 200ug of total protein. Quantification of the protein was detected by measuring the optical density against a standard curve according to the kit specifications.

### 2.8. Statistical Analyses

All statistical analyses were performed in GraphPad Prism 9.5.1 (La Jolla, CA, USA). The Shapiro–Wilk test was performed to assess the data normality, and the Grubbs method, using an alpha set to 0.05, was used to identify and remove outliers. A one-way ANOVA test followed by a post hoc Bonferroni test or Tukey’s test, or a Kruskal–Wallis Test followed by a post hoc Dunn’s test, were used for multiple comparisons as appropriate. Pairwise comparisons were made between each cohort and the controls and the untreated TNBS cohort. An alpha value <0.05 was considered statistically significant.

## 3. Results

### 3.1. Colonic miR146a Gene Expression Was Reduced Following TNBS Enema

Two days following the induction of colitis with TNBS enema, the mice exhibited a significant decrease in their miR146a expression in their colons compared to the control mice (*p* = 0.038, one-way ANOVA with Tukey’s post hoc). Concurrently, the mice that received the TNBS enema showed increased IL-6 and TNF expressions in their colons, which continued to rise to levels of significance by day 5 when compared to the control mice (*p* = 0.003 and *p* < 0.001, respectively, one-way ANOVA with Tukey’s post hoc). The expression of miR146a in the colons of mice induced with TNBS colitis appeared to rise from day 2 to day 5, although not statistically significantly (Figure 1).

### 3.2. CNP-miR146a Enemas Increased Colonic miR146a Expression in Mice Induced with TNBS Colitis

The mice induced with TNBS colitis that received a therapeutic enema on day two with cellulose gel with CNP-miR146a demonstrated a significant rise in their miR146a expression in their colons twelve hours following enema administration compared to the mice that received no therapeutic enema (*p* = 0.029 one-way ANOVA with Tukey’s post hoc). There was also a significant rise in the miR146a expression seen in the colons of mice that received cellulose gel with CNP-miR146a between hour six and twelve after enema delivery (*p* = 0.047, one-way ANOVA with Tukey’s post hoc) (Figure 2).

### 3.3. CNP-miR146a Rescued Weight Loss in Mice with TNBS Colitis

The mice in the untreated cohort, cellulose gel treated cohort, and cellulose gel with CNP-miR146a cohort all lost weight following TNBS administration on day zero. This was demonstrated by a significant increased percentage of weight loss in all three cohorts when compared to the control mice on days one and two (*p* < 0.001 one-way ANOVA with Bonferroni post hoc). The TNBS mice that received cellulose gel with CNP-miR146a on day two demonstrated stabilization in their weight compared to the untreated TNBS mice, starting as early as 24 h after the therapeutic enema on day three and reaching significance at days four and five (*p* = 0.049 and *p* = 0.024, untreated vs. cellulose gel + CNP-miR146a, one-way ANOVA with Bonferroni post hoc). The untreated TNBS mice continued to lose weight compared to the control mice on days four and five (*p* < 0.001 and *p* < 0.001, one-way ANOVA with Bonferroni post hoc), as did the TNBS mice that received cellulose gel enemas (*p* < 0.001 and *p* = 0.002, one-way ANOVA with Bonferroni post hoc). The control mice maintained their weights throughout all five days of the study (Figure 3).

### 3.4. CNP-miR146a Reduced Inflammatory Infiltrates and Erosions on Colon Histology

The colons of the untreated TNBS mice demonstrated significant destruction of the crypt architecture and epithelial erosion, as well as a marked increase in eosinophilia on their histology at the end of five days. The colons of the TNBS mice that received cellulose gel with CNP-miR146a showed reduced inflammatory infiltration and architectural distortion compared to the untreated mice, and in some cases, appeared equivalent to the colons from the control mice (Figure 4). Both the untreated TNBS mice and TNBS mice that received cellulose gel showed significantly higher inflammation scores than the controls (*p* < 0.001, *p* = 0.013, respectively, Kruskal–Wallis with Dunn’s post hoc), whereas the TNBS mice that received cellulose gel with CNP-miR146a showed no significant difference from the controls. The TNBS mice treated with cellulose gel with CNP-miR146a demonstrated significantly lower inflammation scores than the untreated TNBS mice (*p* = 0.006, Kruskal–Wallis with Dunn’s post hoc) (Figure 5).

### 3.5. CNP-miR146a Reduced Proinflammatory Gene Expression in the Colon

Five days after the induction of colitis, untreated TNBS mice demonstrated significantly increased relative gene expression of IL-6 in their colons compared to the controls (*p* = 0.004). The TNBS mice that received cellulose gel on day 2 showed no significant difference in IL-6 expression compared to the untreated TNBS mice. The TNBS mice that received cellulose gel with CNP-miR146a on day two showed reduced IL-6 expression in their colons compared to the untreated TNBS colitis mice (*p* = 0.032) and were not significantly different from the controls. Similarly, five days after the induction of colitis, the untreated TNBS mice demonstrated a significantly increased relative gene expression of TNF in their colons compared to the controls (*p* < 0.001). There was a significant difference in the colonic TNF expression in the TNBS mice that received cellulose gel alone and those that received cellulose gel with CNP-miR146a on day 2 (*p* = 0.013, *p* < 0.001, respectively) (Figure 6).

### 3.6. CNP-miR146a Reduced Protein Concentrations of Proinflammatory Cytokines IL-6 and TNF in the Colon

The protein quantification of IL-6 and TNF by ELISA demonstrated significant increases in IL-6 (*p* = 0.002, one-way ANOVA with Bonferroni post hoc) and TNF (*p* = 0.025, Kruskal–Wallis with Dunn’s post hoc) in the colons of the untreated TNBS mice five days after TNBS instillation compared to the controls. The TNBS mice that received cellulose gel with CNP-miR146a on day two showed decreased IL-6 (*p* = 0.007, one-way ANOVA with Bonferroni post hoc) and TNF (*p* = 0.048, Kruskal–Wallis with Dunn’s post hoc) concentrations compared to the untreated TNBS mice. The TNBS mice that received cellulose gel on day two showed a reduction in IL-6 compared to the untreated TNBS mice (*p* = 0.011, one-way ANOVA with Bonferroni post hoc). There was no difference in the IL-6 or TNF protein concentrations between the TNBS colitis mice treated with CNP-miR146a and the control mice (Figure 7).

### 3.7. CNP-miR146a Enemas Reduced Expression of iNOS2 in TNBS-Injured Colonic Tissue

The untreated TNBS mice and TNBS mice that received cellulose gel enemas demonstrated significantly increased expressions of iNOS2 in their colons compared to the controls (*p* = 0.022 and *p* = 0.005, respectively, Kruskal–Wallis with Dunn’s post hoc) at the end of five days. Treatment with cellulose gel with CNP-mi146a appeared to decrease iNOS2 expression, although not statistically significantly. There was no significant difference between the iNOS relative expression in the colons of the controls versus the TNBS mice treated with cellulose gel with CNP-miR146a (Figure 8).

## 4. Discussion

In this study, we show that a single dose of 25 ng of CNP-miR146a delivered as an enema improved clinical disease in a TNBS murine model for acute colitis, as demonstrated by improvements in weight loss and colon inflammation as seen on histology. The delivery of cellulose gel with CNP-miR146a enemas was associated with a rise in miR146a expression at 12 h following enema delivery and the downregulation of proinflammatory cytokines IL-6 and TNF at 72 h following delivery. These results are consistent with the expected timeline of transcriptional changes resulting from miR146a downregulating the NF-kB pathway. Concurrently, we showed the downregulation of iNOS2, a potent ROS producer that is upregulated by the NF-kB pathway.

The TNBS acute colitis model is a widely used animal model for studying acute colitis that resembles acute Crohn’s colitis in humans due to the predominant Th1 immune response it elicits in the colon. In the first 48 h following TNBS administration, mice experience weight loss, which can be severe, usually due to a combination of diarrhea, poor colonic fluid absorption, and reduced appetite. The diarrhea-associated weight loss and inflammatory cell infiltration of the colon seen in the TNBS model are hallmarks of IBD in humans [7]. After 48 h, mice in the TNBS model continue to lose weight as a consequence of a systemic inflammatory response [32], which resembles the sustained inflammatory signaling seen in human IBD. These processes were evident in the steady rise of IL-6 and TNF gene expression seen in the colons of mice on two and five days following TNBS enema instillation.

The regional delivery of CNP-miR146a improved weight loss and decreased colon inflammation in mice in this model, as seen on histology and through the downregulation and normalization of proinflammatory cytokines IL-6 and TNF. The downregulation of IL-6 and TNF in the colon has significant clinical implications. IL-6 plays a critical role in differentiating naïve CD4+ T-cells into Th17 cells, in addition to activating other proinflammatory pathways, like the signal transducer and activator of transcription (STAT) 3 pathway [4,5,36]. TNF is one of the major cytokines secreted by Th1-activated immune cells and can cause tissue destruction by triggering cellular apoptosis through NF-κB activation [4,37]. Th1 and Th17 are proinflammatory subtypes of CD4+ T lymphocytes and predominantly mediate inflammation in Crohn’s disease [7]. In physiological conditions, Th1 and Th17 cells play important roles in the adaptive immune system by eliminating pathogens directly and indirectly by recruiting macrophages and CD8+ effector T cells [5,38]. Excessive Th1 and Th17 responses are part of the pathogenicity of IBD, and increased levels of activated Th1 and Th17 cells have been found in the inflamed mucosa of both UC and Crohn’s disease patients [39].

The reduced immune cell infiltration seen on histology of CNP-miR146a-treated TNBS mice may be explained by the downregulation of Th17 differentiation and Th1 activation through reduced IL-6 and TNF expression. In this study, colons from the untreated TNBS mice showed increased architectural distortion and destruction of epithelial crypts, as well as a marked influx of immune cells into all tissue layers. This was also true for the TNBS mice treated with cellulose gel enema. Meanwhile, the TNBS mice treated with cellulose gel with CNP-miR146a demonstrated immune cell infiltration limited to mostly the mucosa, suggesting that our treatment may have prevented deeper tissue infiltration by halting the inflammatory process.

Interestingly, the TNBS mice treated with cellulose gel alone demonstrated isolated reductions in the IL-6 concentrations and TNF gene expression in their colons. The cellulose gel used in this study is a derivative of hydroxyethyl cellulose. Hydroxyethyl cellulose is a water-soluble biocompatible polymer with a well-established safety profile, low toxicity, and non-immunogenicity, and it has previously been used for small-molecule drug delivery to mucosal surfaces [40,41,42,43]. Hydroxyethyl cellulose gel was used as a vehicle for CNP-miR146a to provide viscosity and prevent immediate leakage following the rectal instillation. The reductions in the proinflammatory cytokines in the colons of the mice that received hydroxyethyl cellulose gel may have resulted from the capacity of hydrogels to absorb physiological exudates, including proinflammatory cytokines [41]. It is also possible that the hydrogel provided a degree of hydration to these mice through the absorption of water from the gel by the inflamed mucosa. It is significant to note that the TNBS mice treated with cellulose gel did not exhibit improvements in weight loss or colonic inflammation on histology compared to the untreated TNBS mice, and only TNBS mice that received cellulose gel with CNP-miR146a showed improvements on both the clinical and molecular levels.

MiR146a expression has been shown to be elevated in the intestinal tissue of acutely diseased patients with IBD [44,45]. This is contrary to what was seen in our study, in which miR146a expression was reduced in the colons of the mice two days following TNBS administration. Few studies have examined miRNA in the TNBS model, and none to our knowledge have specifically investigated miR146a. The decrease in miR146a expression seen in our study may reflect the acute nature of the TNBS model, which creates intense and severe colitis immediately following enema. This disease pattern prevents animals from developing any compensatory mechanisms or adaptations that might otherwise be responsible for an increased production of miR146a in human IBD, which is a chronic inflammatory disease. This is supported by the data in our study that show a trend toward increased miR146a expression in the colons of mice between two and five days following TNBS administration. Despite the relative increase in miR146a, the untreated mice continued to exhibit increased expressions of IL-6 and TNF on day five.

In this regard, it is possible that increasing miR146a in the colon through the delivery of CNP-miR146a can augment the body’s natural response to inflammation. Given that miR-146a is an anti-inflammatory miRNA, it is reasonable that its production should increase in response to inflammation, however, its relative increase in the colons of IBD patients is likely insufficient in countering the overwhelming inflammation that is propagated through multiple signaling pathways. After all, the NF-κB pathway can be activated through mechanisms other than IRAK1 and TRAF6, and several other signaling pathways, like STAT3, are involved in the inflammation and immune response of IBD [5]. In this study, by administering our therapeutic 48 h following TNBS administration, we were able to augment miR146a levels at the point of maximum inflammation.

The therapeutic effects of CNP-miR146a in this study are likely a consequence of both the anti-inflammatory effects of miR-146a and the antioxidant effects of CNP [46]. The chemical conjugation of CNP to miR146a stabilizes the miRNA for therapeutic delivery and provides synergistic effects to the anti-inflammatory properties of miR146a by providing antioxidant radical scavenging capabilities [47]. The ability of CNP to sequester ROS provides a secondary mechanism of modulating inflammatory signaling [33]. In this study, the TNBS mice that were untreated or received cellulose enemas demonstrated significant upregulation of iNOS in their colons at the end of five days. This is consistent with data that have shown an upregulation of iNOS in the inflamed colons of patients with IBD [48]. The relative decrease in NOS expression in the colons of mice that received treatment with cellulose with CNP-miR146a indirectly suggests decreased ROS production. Since iNOS can be induced by NF-κB, it is difficult to determine whether this effect was a direct consequence of the antioxidant properties of CNP or of the downregulation of NF-κB by miR146a. Most likely, both elements are contributory, as has been shown in our previous studies.

The local delivery of CNP-miR146a minimizes the potential for systemic absorption and undesired side effects. The delivery of miRNA as a therapeutic is challenging given the unstable nature of miRNA and their susceptibility to degradation [49]. One prior study attempted the delivery of miR146a to rats with TNBS colitis through the intravenous administration of a miR146a using a lentivirus vehicle [50]. Unfortunately, the use of lentivirus as a vehicle for drug delivery is not practical in a clinical setting. Additionally, since current treatments for IBD most often require intravenous administration [17], we sought to investigate the therapeutic administration of CNP-miR146a for IBD through regional delivery. To our knowledge, only one previous study has discussed the regional delivery of therapeutic miRNA as an enema in IBD mouse models [46].

The results of our study must be interpreted with the consideration of important limitations. Only male mice were used due to sex-associated differences in the severity of TNBS colitis [31]. Additionally, while the TNBS model does not use artificially created genetic deficiencies or mutations that do not exist in human IBD, it does depend on the instillation of a foreign chemical. The delivery of an enema of TNBS creates a more focal injury in the distal colon, as was evident on the gross pathology and histology. This does not represent the full spectrum of Crohn’s disease, which often involves the proximal colon and small bowel. Additionally, this model represents an acute form of colitis, and the results cannot be extrapolated to chronic models of disease. Further studies will be needed to determine the efficacy of CNP-miR146a in chronic disease models of other parts of the intestine.

## 5. Conclusions

The local delivery of CNP-miR146a via enema in a TNBS mouse model of acute colitis significantly decreases inflammatory signaling, resulting in improved clinical disease, with decreased weight loss and reduced inflammatory infiltration of the colon. The local delivery of CNP-miR146a also decreases the potential complications associated with systemic immunosuppression. CNP-miR146a may mitigate the effects of an overactivated immune system by upregulating anti-inflammatory miR146a and decreasing ROS. CNP-miR146a shows promising therapeutic potential for use in IBD. Further studies are necessary to characterize the mechanism of action of CNP-miR146a and its applicability in chronic animal models of IBD.

## Figures and Tables

**Figure 1 nanomaterials-14-00894-f001:**
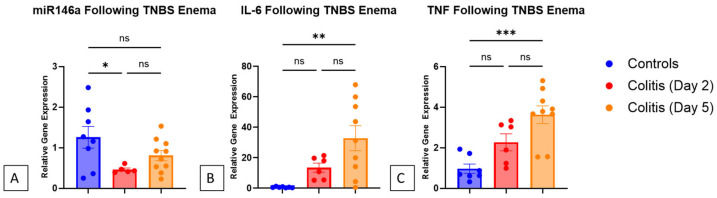
(**A**) The mice induced with colitis via TNBS enema demonstrated a significant decrease in miR146a expression in their colons compared to the control mice (*p* = 0.038) two days following TNBS administration. The expression of miR146a increased in the colons of mice with TNBS colitis by day 5. (**B**) Following TNBS enema, the expression of IL-6 gradually rose, with a significant increase seen in the controls on day 5 (*p* = 0.003). (**C**) TNF expression followed the same pattern, showing a steady rise and significant increase in the controls on day 5 (*p* < 0.001). The asterisks indicate statistical significance using one-way ANOVA with Tukey’s post hoc, and the bars indicate the standard error of the mean.

**Figure 2 nanomaterials-14-00894-f002:**
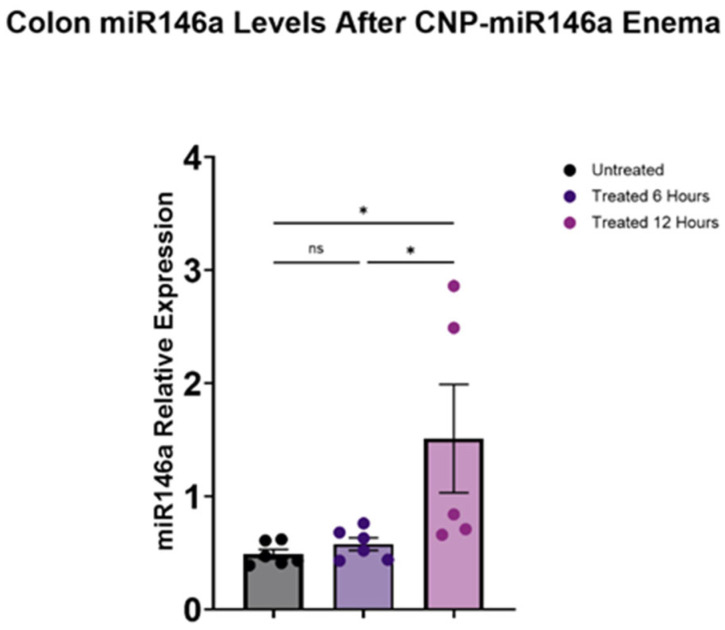
The mice induced with TNBS colitis that received cellulose gel with CNP-miR146a enemas on day two were euthanized at six and twelve hours following the therapeutic enema and compared to the untreated TNBS mice. The MiR146a expression in the colons of the TNBS mice was significantly increased at twelve hours following the delivery of therapeutic cellulose gel with CNP-miR146a enema (pink) compared to the colons of the untreated mice (black) (*p* = 0.029). There was a significant rise in the colonic miR146a expression between six (purple) and twelve hours after cellulose gel with CNP-miR146a enema delivery (*p* = 0.047). The asterisks indicate statistical significance using one-way ANOVA with Tukey’s post hoc, and the bars indicate the standard error of the mean.

**Figure 3 nanomaterials-14-00894-f003:**
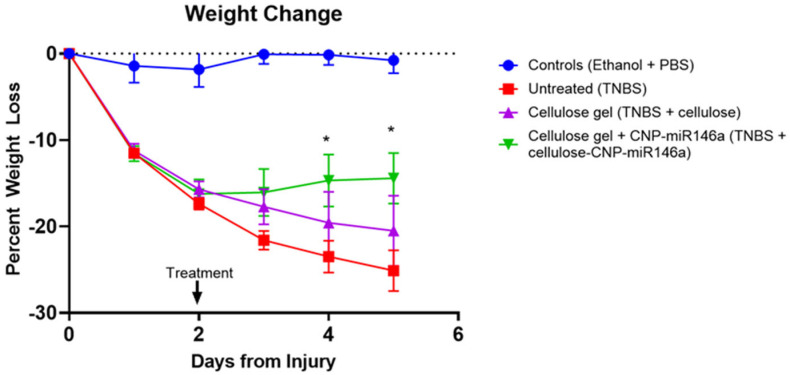
The control mice (blue) maintained their weight across the five-day experiment. The mice that received TNBS without any subsequent treatment (red) lost weight immediately following TNBS instillation on day 0 and continued to lose weight throughout all 5 days of the experiment. The mice that received TNBS on day 0 and cellulose gel enema on day 2 (purple) also showed sustained weight loss throughout all 5 days. The mice that received TNBS on day 0 and cellulose gel with CNP-miR146a on day 2 demonstrated stabilization in their weight starting on day 3, reaching statistical significance on days 4 and 5 (*p* = 0.049 and *p* = 0.024, respectively). The asterisks indicate statistical significance using one-way ANOVA with Bonferroni post hoc, and the bars indicate the standard error of the mean.

**Figure 4 nanomaterials-14-00894-f004:**
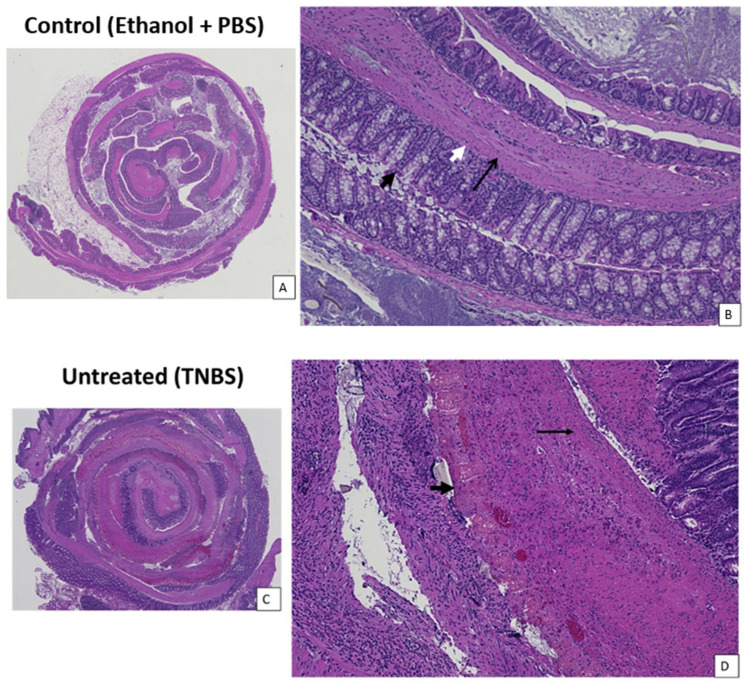
Representative images of colon histology on brightfield microscopy with hematoxylin and eosin (H&E) staining shown at 20× (left) and 100× magnification (right) five days after TNBS instillation. (**A**,**B**) A control colon with normal-appearing mucosa (thick black arrow) and abundant goblet cells seen as clear vesicles and distinct layers of the muscularis propria—the circular muscle (white arrow) and the longitudinal muscle (thin black arrow). (**C**) An untreated TNBS colon demonstrating inflammation throughout the colon, with areas of obliterated mucosa and significant eosinophilia represented by areas of dark pink. (**D**) An untreated TNBS colon showing a section of obliterated mucosa (thick black arrow) with loss of goblet cells and epithelial architecture. Muscularis propria is thickened and contains full-thickness inflammatory infiltration (thin black arrow). (**E**) A colon from a TNBS mouse that received cellulose gel, demonstrating similar inflammation throughout the colon with (**F**) a close-up showing similar architectural destruction of the mucosa (thick black arrow) with goblet cell loss and the penetration of inflammatory cells into the submucosa (white arrow) and the muscularis propria (thin black arrow). (**G**) A colon of a TNBS mouse that received cellulose gel with CNP-miR146a showing reduced mucosal architectural destruction and inflammatory infiltration with (**H**) intact mucosa and goblet cells (thick black arrow) without architectural destruction and inflammatory infiltrates in the submucosa (white arrow). Although some regions of the colon had inflammatory infiltration to the muscularis propria, many areas appeared normal with distinct circular and longitudinal muscle layers (thin black arrow).

**Figure 5 nanomaterials-14-00894-f005:**
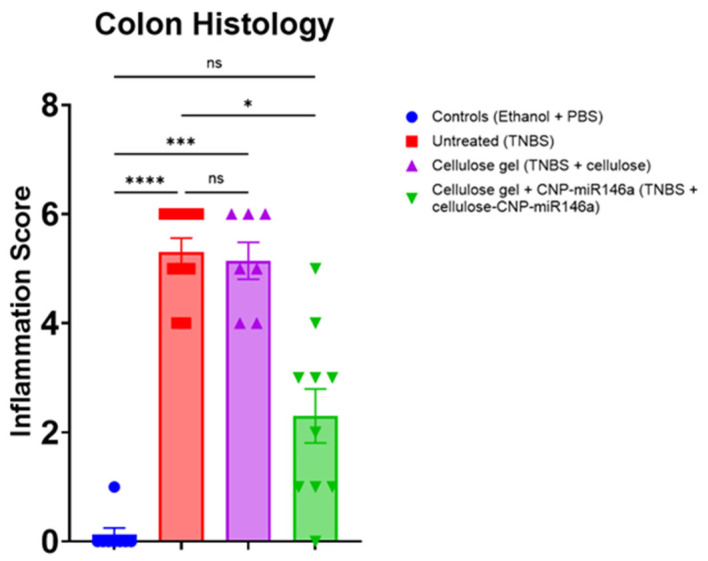
The untreated TNBS mice (red) and TNBS mice that received cellulose gel (purple) exhibited higher inflammation scores on their histology compared to the controls (blue) (*p* < 0.001 and *p* < 0.001, respectively). The colons from the TNBS mice treated with cellulose gel with CNP-miR146a (green) demonstrated significantly decreased inflammation scores when compared to the untreated TNBS mice (*p* = 0.018). The asterisks indicate statistical significance using Kruskal–Wallis with Dunn’s post hoc, and the bars indicate the standard error of the mean.

**Figure 6 nanomaterials-14-00894-f006:**
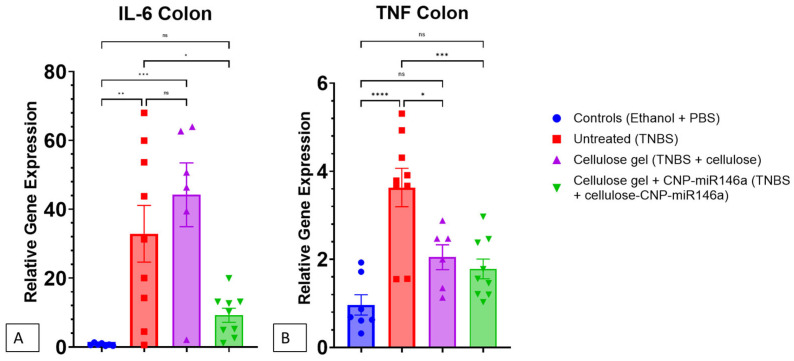
The relative gene expressions of IL-6 (**A**) and TNF (**B**) in the colon 5 days following TNBS instillation and 3 days following treatment enema. (**A**) The untreated TNBS mice (red) and the TNBS mice that received cellulose gel (purple) demonstrated significant increases in IL-6 compared to the control mice (*p* = 0.004 and *p* < 0.001, respectively). The TNBS mice that received cellulose gel with CNP-miR146a (green) demonstrated a significant decrease in IL-6 gene expression compared to the untreated TNBS mice (*p* = 0.032). (**B**) Similarly, the untreated TNBS mice exhibited increased expression of TNF in their colons (*p* < 0.001, controls vs. untreated TNBS) and the TNBS mice that received cellulose gel with CNP-miR146a showed a reduction of TNF (*p* < 0.001, cellulose gel + CNP-miR146a vs. untreated TNBS). The asterisks indicate statistical significance using one-way ANOVA with Bonferroni post hoc, and the bars indicate the standard error of the mean.

**Figure 7 nanomaterials-14-00894-f007:**
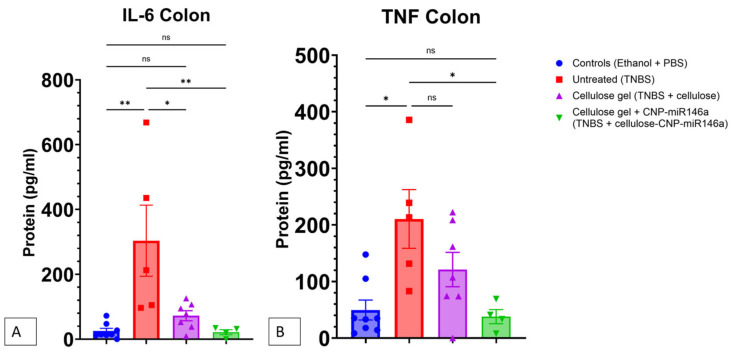
IL-6 (**A**) and TNF (**B**) protein concentrations in the colon 5 days following TNBS instillation and 3 days following the treatment enema measured by ELISA. The untreated TNBS mice (red) showed increased levels of IL-6 in their colons compared to the controls (blue) (*p* = 0.002, one-way ANOVA with Bonferroni post hoc) and increased levels of TNF although not significant (*p* = 0.025, Kruskal–Wallis with Dunn’s post hoc). The colons of the TNBS mice that received CNP-miR146a (green) demonstrated decreased IL-6 and TNF levels relative to the untreated TNBS mice (*p* = 0.007 and *p* = 0.048, respectively). The TNBS mice that received cellulose gel only (purple) demonstrated reduced IL-6 content in their colons compared to the untreated TNBS mice (*p* = 0.011). The asterisks indicate statistical significance, and the bars indicate the standard error of the mean.

**Figure 8 nanomaterials-14-00894-f008:**
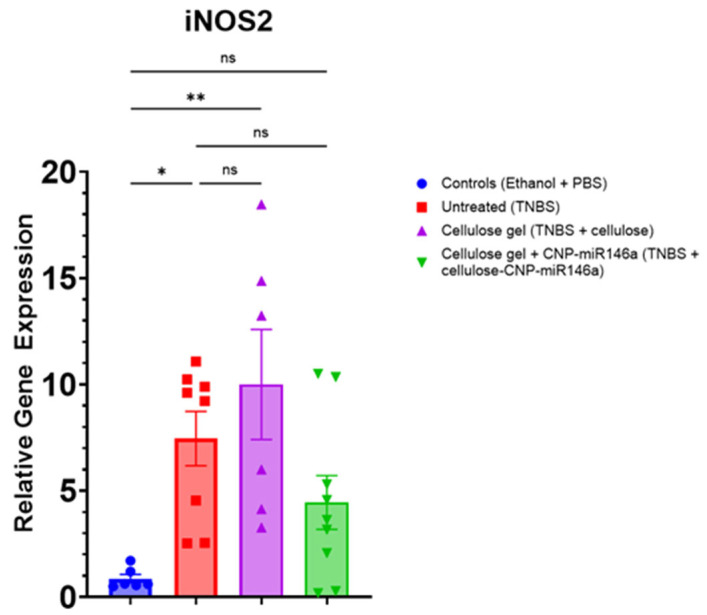
Relative gene expressions of inducible nitric oxide synthase 2 (iNOS2) in the colon 5 days following TNBS administration and 3 days following treatment. The untreated TNBS mice (red) demonstrated a significant increase in iNOS2 expression in their colons compared to the control mice (blue) (*p* = 0.022). The TNBS mice that received cellulose gel enemas on day 2 (purple) also demonstrated an increased expression of iNOS2 relative to the controls (*p* = 0.005). The TNBS mice treated with cellulose gel with CNP-miR146a (green) demonstrated reduced levels of iNOS2 expression relative to the untreated TNBS and the TNBS with cellulose groups. There was no significant difference in iNOS2 expression between the TNBS mice treated with cellulose with CNP-miR146a and the controls. The asterisks indicate statistical significance using Kruskal–Wallis with Dunn’s post hoc, and the bars indicate the standard error of the mean.

**Table 1 nanomaterials-14-00894-t001:** Scoring system used for evaluation of colon histology. Reprinted/adapted with permission from Ref. [35]. Copyright 2014, IJCEP.

Inflammatory Cell Infiltrate		Score 1	Intestinal Architecture		Score 2
Severity	Extent		Epithelial Changes	Mucosal Architecture	
Mild	Mucosa	1	Focal erosions		1
Moderate	Mucosa and submucosa	2	Erosions	±Focal ulcerations	2
Marked	Transmural	3		Extended ulcerations ± granulation tissue ± pseudopolyps	3
			Sum of scores 1 and 2		0–6

## Data Availability

The raw data supporting the conclusions of this article will be made available by the authors upon request.

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
