# Peer review of "Targeting Inflammation and Oxidative Stress to Improve Outcomes in a TNBS Murine Crohn’s Colitis Model"

_nanomaterials, 2024, doi:10.3390/nano14100894_

Round 1

Reviewer 1 Report

Comments and Suggestions for Authors

The paper reports on the anti-inflammatory action of cerium oxide nanoparticles (CNP) conjugated to microRNA 146a in a model for Crohn’s disease. The results obtained confirm promising therapeutic potential of this hybrid nanomaterial in the treatment of colitis. The subject of the paper fits well the scope of Nanomaterials journal.

I have the following comments:

1. Cerium oxide nanoparticles conjugated to micro-RNA could hardly be referred to as novel compound, this is rather a composite nanomaterial or a hybrid material.

2. Please provide quantitative data on the micro-RNA to CNPs ratio in the composite material.

3. What is the nature of the interaction between micro-RNA and CNPs? Is it just physisorption or no? Please explain.

4. Could the effects observed be due to the local increase in micro-RNA concentration due to its sorption on the surface of CNPs? Please comment.

5. I would suggest adding more recent references concerning antioxidant and enzyme-like activity of CNPs.

Reviewer 2 Report

Comments and Suggestions for Authors

The paper entitled “Targeting Inflammation and Oxidative Stress to Improve Outcomes in a TNBS Murine Crohn’s Colitis Model” by Apte and colleagues describes an interesting study focusing on the potential application of their previously developed cerium oxide nanoparticles coupled to miR146a in the treatment of Crohn’s disease. The paper is well organized and well written, and results are clearly presented. I think it is a good quality work, but I have few suggestions for the Authors.

1)      Please correct “Turkey’s post-hoc test”; it is “Tukey’s”.

2)      In figure 3, it is clearly evident that the difference between the weight loss in animals receiving the cellulose gel and in animals receiving the cellulose gel + CNP-miR146a is not significant; on lines 470-473 within the Discussion section the Authors state that only mice receiving cellulose gel + CNP-miR146a showed a clinical improvement, with a partial weight gain. Actually, this is true, but I think the Authors should add a comment regarding the lack of significance between the data related to weight loss in mice receiving the plain gel and the mice receiving the CNP-miR146a loaded gel.

3)      In figure 4, the panels are arranged in a strange way, with panels C, D, E and F preceding the A and B ones. This is quite confusing when reading the caption and observing the images, so I think the Authors should move the images (or change the panels letters) in order to have panels that follows the alphabetical order.
